# Structural basis of liprin-α-promoted LAR-RPTP clustering for modulation of phosphatase activity

Xingqiao Xie[1,2], Ling Luo[1], Mingfu Liang[1], Wenchao Zhang[1], Ting Zhang[1,2], Cong Yu[1,3] & Zhiyi Wei [1,2]*

Leukocyte common antigen-related receptor protein tyrosine phosphatases (LAR-RPTPs) are cell adhesion molecules involved in mediating neuronal development. The binding of LAR-RPTPs to extracellular ligands induces local clustering of LAR-RPTPs to regulate axon growth and synaptogenesis. LAR-RPTPs interact with synaptic liprin-α proteins via the two cytoplasmic phosphatase domains, D1 and D2. Here we solve the crystal structure of LAR_D1D2 in complex with the SAM repeats of liprin-α3, uncovering a conserved two-site binding mode. Cellular analysis shows that liprin-αs robustly promote clustering of LAR in cells by both the liprin-α/LAR interaction and the oligomerization of liprin-α. Structural analysis reveals a unique homophilic interaction of LAR via the catalytically active D1 domains. Disruption of the D1/D1 interaction diminishes the liprin-α-promoted LAR clustering and increases tyrosine dephosphorylation, demonstrating that the phosphatase activity of LAR is negatively regulated by forming clusters. Additionally, we find that the binding of LAR to liprin-α allosterically regulates the liprin-α/liprin-β interaction.

[1] Department of Biology, Southern University of Science and Technology, Shenzhen, Guangdong 518055, China. [2] Academy for Advanced Interdisciplinary Studies, Southern University of Science and Technology, Shenzhen, Guangdong 518055, China. [3] Guangdong Provincial Key Laboratory of Cell Microenvironment and Disease Research, and Shenzhen Key Laboratory of Cell Microenvironment, Shenzhen, Guangdong 518055, China. *email: weizy@sustech.edu.cn

The leukocyte common antigen-related receptor protein tyrosine phosphatases (LAR-RPTPs), also named as Type IIα RPTPs, are cell surface receptors important for the developing nervous system[1–3]. By participating in both cell–cell and cell–extracellular matrix (ECM) adhesions, LAR-RPTPs play critical roles in mediating axon guidance, neurite outgrowth, as well as synapse formation and differentiation[4–8]. Mutations in human LAR-RPTP coding genes have been found associated with neurological disorders, such as autism, schizophrenia, and bipolar disorder[9–13]. The LAR-RPTP family consists of three members in vertebrates, LAR, PTP-σ, and PTP-δ, each containing three immunoglobulin-like domains and several fibronectin III-like domains in the N-terminal extracellular part and two tyrosine phosphatase domains, a catalytically active D1 domain and a catalytically inactive D2 domain, in the C-terminal cytoplasmic part[14,15] (Fig. 1a). The phosphatase activity of LAR-RPTPs is required for proper neuronal signaling and development. With a cysteine-to-serine mutation in the catalytic site of the D1 domain, the catalytically inactive LAR-RPTP failed to control the rate of axon growth[16] or promote synapse formation and function[17]. However, little is known about the activity regulation mechanism of LAR-RPTPs.

Like other cell adhesion molecules, LAR-RPTPs use their large extracellular regions to interact with a variety of extracellular ligands, including proteoglycans and postsynaptic adhesion proteins[6,7,18]. The extracellular interactions often trigger local clustering of LAR-RPTP in axon growing tips and presynaptic terminals for guided axon growth and presynaptic differentiation, found by studies on the LAR-RPTP ectodomain structures in complex with heparan sulfate and chondroitin sulfate proteoglycans (HSPG and CSPG)[19], Slit- and Trk-like protein 1 (Slitrk1)[20], and IL-1 receptor accessory protein-like 1 (IL1RAPL1)[21]. However, the downstream effects of LAR-RPTP clustering remain elusive.

Liprin-α proteins, having four vertebrate paralogues, liprin-α1–α4[22,23], are well-known for their scaffolding function in synaptic development and activity[24–28]. As the intracellular effectors of LAR-RPTP, the liprin-α proteins interact with the catalytically inactive D2 domain of LAR-RPTP via their highly conserved C-terminal three sterile alpha motif (SAM) domains[22,29] (Fig. 1a and Supplementary Fig. 1). The binding of liprin-α to LAR-RPTP promotes spine formation and presynaptic assembly and differentiation[17,30,31]. The three SAM domains (SAM123) in liprin-α fold together to form a versatile protein-binding module[32]. In addition to LAR-RPTP binding, liprin-α_SAM123 also binds to calcium/calmodulin-dependent serine protein kinase (CASK)[32,33], liprin-β[22], and mammalian synapse-defective-1 (mSYD1)[34].

To explore the molecular basis for the liprin-α-mediated function of LAR, we solved the crystal structure of the liprin-α3_SAM123/LAR_D1D2 complex. The complex structure reveals two conserved binding sites between the first two SAM domains in liprin-α3 and the D2 domain in LAR. In COS7 cells, the binding of liprin-α to LAR promotes the formation of LAR clusters at ventral plasma membranes. Interestingly, disruption of the packing interface between two adjacent LAR D1 domains found in the crystal dramatically decreased the liprin-α-promoted cluster formation. Further structural and biochemical analysis indicates that the D1/D1 packing interferes with the phosphatase activity of LAR by blocking the substrate-binding pocket, providing mechanistic insights into understanding the functional linkage of the liprin-α/LAR-RPTP interaction, clustering, and phosphatase activity. Surprisingly, although the binding surfaces of LAR and liprin-β on liprin-α_SAM123 do not overlap with each other, LAR can compete with the binding of liprin-β to liprin-α through an allosteric way.

## Results

**Biochemical characterization of the liprin-α/LAR-RPTP interaction.** To understand the molecular mechanism underlying the liprin-α/LAR-RPTP interaction, we purified the C-terminal SAM repeats of liprin-α2 (α2_SAM123) and the cytoplasmic D1 and D2 domains of LAR (LAR_D1D2), which are highly conserved across species (Fig. 1a and Supplementary Figs. 1 and 2). As expected, α2_SAM123 and LAR_D1D2 formed a complex in solution and the binding affinity between these two fragments was measured as 3.4 μM (Fig. 1b, c). However, neither the SAM1 domain only nor the SAM23 tandem showed a detectable binding to LAR_D1D2 (Fig. 1d, e and Table 1), indicating that the three SAM repeats all contribute to the target binding by forming a structural supramodule[32]. Likewise, the D2 domain alone, the previously mapped binding region for liprin-α[29], was not sufficient to maintain the full binding capacity of LAR to liprin-α (Table 1), consistent with a recent study of the liprin-α/PTPσ interaction[35]. To probe the potential LAR-binding sites in liprin-α, we prepared several deletion mutants of α2_SAM123, including

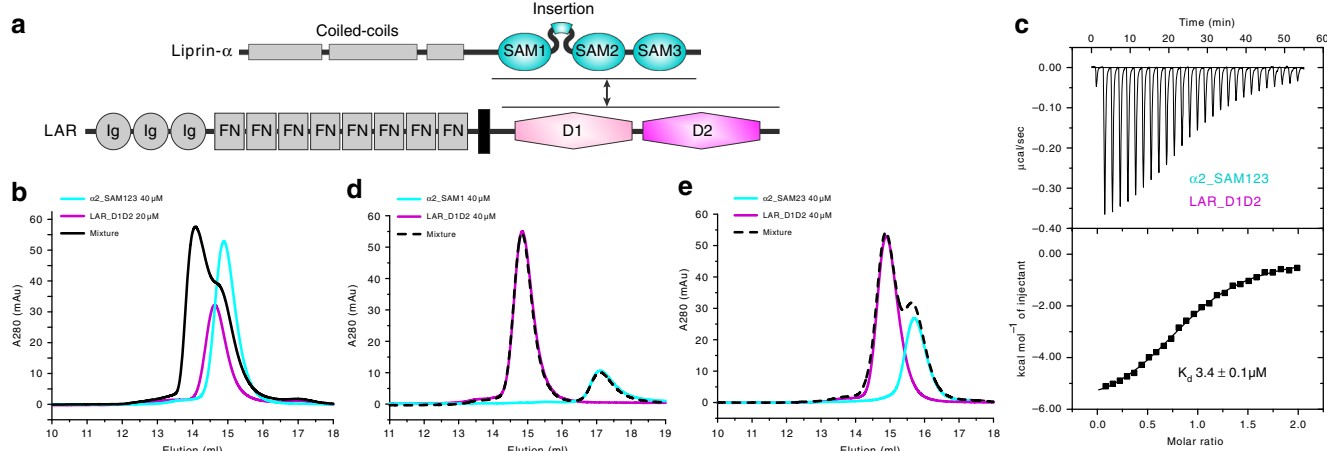

**Fig. 1 Biochemical characterization of the liprin-α and LAR-RPTP interaction. a** Cartoon diagrams of domain organizations of liprin-α and LAR. The color-coding of the regions is applied throughout the entire manuscript except as otherwise indicated. **b, d, e** Analytical gel filtration analysis showing the interaction with LAR requires all three SAM domains of liprin-α. **c** Isothermal titration calorimetry (ITC)-based measurement of the binding of α2_SAM123 to LAR_D1D2.

**Table 1 Summary of binding affinities measured by ITC between various fragments of liprin-αs and LAR-RPTPs.**

| Liprin-α | LAR-RPTP | $K_d$ (μM) |
|---|---|---|
| α2_SAM123 | LAR_D1D2 | 3.4 ± 0.1 |
| α2_SAM1 | LAR_D1D2 | Not detectable |
| α2_SAM23 | LAR_D1D2 | Not detectable |
| α2_SAM123 | LAR_D2 | 14 ± 1 |
| α2_SAM123ΔN | LAR_D1D2 | 12 ± 1 |
| α2_SAM123ΔInsertion | LAR_D1D2 | 3.0 ± 0.4 |
| α2_SAM123ΔInsertion | PTPR-σ_D1D2 | 2.4 ± 0.3 |
| α2_SAM123ΔInsertion | PTPR-δ_D1D2 | 6.4 ± 1.5 |
| α1_SAM123 | LAR_D1D2 | 3.8 ± 0.8 |
| α3_SAM123 | LAR_D1D2 | 3.6 ± 0.2 |
| α4_SAM123 | LAR_D1D2 | 5.9 ± 0.6 |

The corresponding titration curves can be found in Fig. 1c, Fig. 2i, and Supplementary Fig. 3

**Table 2 Data collection and refinement statistics.**

| Data collection | |
|---|---|
| Space group | C2 |
| *Cell dimensions* | |
| a, b, c (Å) | 250.378, 147.678, 143.972 |
| α, β, γ (°) | 90.000, 103.852, 90.000 |
| Resolution range (Å) | 50.00–2.85 (2.90–2.85) |
| $R_{merge}$ (%)[a] | 15.0 (106.8) |
| $CC_{1/2}$[b] | (0.744) |
| $I/\sigma(I)$ | 14.0 (1.5) |
| Completeness (%) | 99.7 (99.9) |
| Redundancy | 8.2 (8.6) |
| *Refinement* | |
| Resolution (Å) | 50–2.85 (2.92–2.85) |
| No. of reflections | 117,634 (7732) |
| $R_{work}/R_{free}$ (%)[c] | 19.2 (27.0)/23.0 (31.0) |
| No. of atoms | |
| Protein | 24,569 |
| Ligand | 103 |
| Water | 161 |
| Mean B (Å$^2$) | |
| Protein | 82.0 |
| Ligand | 94.7 |
| Water | 48.8 |
| R.m.s deviations | |
| Bond lengths (Å) | 0.007 |
| Bond angles (°) | 0.922 |
| Ramachandran analysis | |
| Favored region (%) | 97.01 |
| Allowed region (%) | 2.73 |
| Outliers (%) | 0.26 |

The numbers in parentheses represent values for the highest resolution shell
[a]$R_{merge} = \sum |I_i - I_m| / \sum I_i$, where $I_i$ is the intensity of the measured reflection and $I_m$ is the mean intensity of all symmetry related reflections
[b]$CC_{1/2}$ is the correlation coefficient of the half datasets
[c]$R_{work} = \Sigma ||F_{obs}| - |F_{calc}|| / \Sigma |F_{obs}|$, where $F_{obs}$ and $F_{calc}$ are observed and calculated structure factors
$R_{free} = \Sigma_T ||F_{obs}| - |F_{calc}|| / \Sigma_T |F_{obs}|$, where $T$ is a test data set of about 1.7 % of the total reflections randomly chosen and set aside prior to refinement

the deletion of the charged sequence in the very N-terminus of the SAM1 domain (α2_SAM123ΔN) and of the insertion loop connecting SAM1 and SAM2 (α2_SAM123ΔInsertion) (Supplementary Fig. 1), and analyzed the binding affinities between these mutants and LAR_D1D2 (Table 1 and Supplementary Fig. 3). Interestingly, α2_SAM123ΔN showed an approximately fourfold decreased binding affinity with LAR_D1D2, suggesting that the charge–charge interaction plays a role in the liprin-α/LAR interaction. In contrast, α2_SAM123ΔInsertion had little effect on the binding affinity. Since the insertion loop is essential for the binding of liprin-α to CASK, our observation demonstrates that liprin-α may interact with LAR and CASK by using different surfaces on SAM123.

Given the functional diversities of the liprin-α and LAR-RPTP family members[36–39], we asked whether the liprin-α/LAR-RPTP interactions have isoform specificity. To address this question, we measured the binding affinities between four liprin-αs and three LAR-RPTPs (Table 1 and Supplementary Fig. 3). As these measurements showed comparable binding affinities, it is likely that the associations between different liprin-αs and LAR-RPTPs share the same binding mode.

**Overall structure of the liprin-α3_SAM123/LAR_D1D2 complex.** To uncover the binding mode between liprin-αs and LAR-RPTPs, we aimed to solve the liprin-α/LAR-RPTP complex structure. We extensively screened crystals by using 12 possible liprin-α_SAM123/LAR-RPTP_D1D2 pairs. Finally, we successfully obtained high quality crystals of the α3_SAM123/LAR_D1D2 complex and determined the crystal structure with a 2.85 Å resolution (Table 2). In one asymmetric unit, four LAR_D1D2 molecules were assigned with essentially the same conformation, which shows a small change of the orientation between the D1 and D2 domains by comparing to the apo structure of LAR_D1D2 (Supplementary Fig. 4a). Except for the insertion loop connecting the SAM1 and SAM2 domains and a few short loops in the SAM3 domain, three α3_SAM123 molecules were modeled (Supplementary Fig. 4b), each interacting with one LAR_D1D2 molecule to form a 1:1 complex with the buried area of ~1100 Å$^2$ on both sides. Due to lacking of clear electron density, the fourth α3_SAM123 molecule was almost unmodeled (Supplementary Fig. 4b). Crystal packing analysis showed that placing a LAR_D1D2-bound α3_SAM123 molecule here, near a crystallographic twofold axis, would generate sever clashes between this α3_SAM123 molecule and its two-fold symmetry-related molecule (Supplementary Fig. 4c), thereby preventing α3_SAM123 from packing stably at this position.

In the complex, LAR interacts with liprin-α3 via the D2 domain via two conserved binding sites, site-I and II (Fig. 2a and Supplementary Fig. 5a). The highly conserved property of the binding interface strongly supports that the binding mode found in the α3_SAM123/LAR_D1D2 complex is likely to be shared by all liprin-αs and LAR-RPTPs (Fig. 2b). Although sharing the similar domain organization with liprin-α[22], liprin-β does not interact with LAR[22,40] (Supplementary Fig. 5b).

The interface residues in site-I interlock mainly through hydrophobic interactions. On one hand, the C-terminal residues of the α3_D2-helix in LAR, particularly Q1828 and F1829, are inserted into a cleft between the SAM1 and SAM2 domains in liprin-α3 (Fig. 2a, c), explaining the observation that the separation of SAM123 to SAM1 and SAM23 abolished the binding of liprin-α to LAR (Fig. 1d, e). We noted with interest that the corresponding residues of Q1828 and F1829 in PTPσ have been recently found in a mutagenesis study to be liprin-α2-binding sequence[35], further supporting the high conservation of the liprin-α/LAR-RPTP binding mode. On the other hand, a Trp residue in the SAM1 domain of liprin-α3, W856 puts its bulky sidechain into a hydrophobic pocket formed by the β8–β10 sheets and the α3_D2-helix in LAR (Fig. 2c and Supplementary Fig. 2). Several hydrogen bonds and salt bridges further strengthen the intermolecular interactions in site-I (Fig. 2c).

Compared to site-I, site-II occupies much smaller surfaces and likely plays a minor role in the liprin-α/LAR interaction. The very N-terminal residues in the αN-helix and its N-terminal loop of the SAM1 domain interact with the pocket mainly formed by the

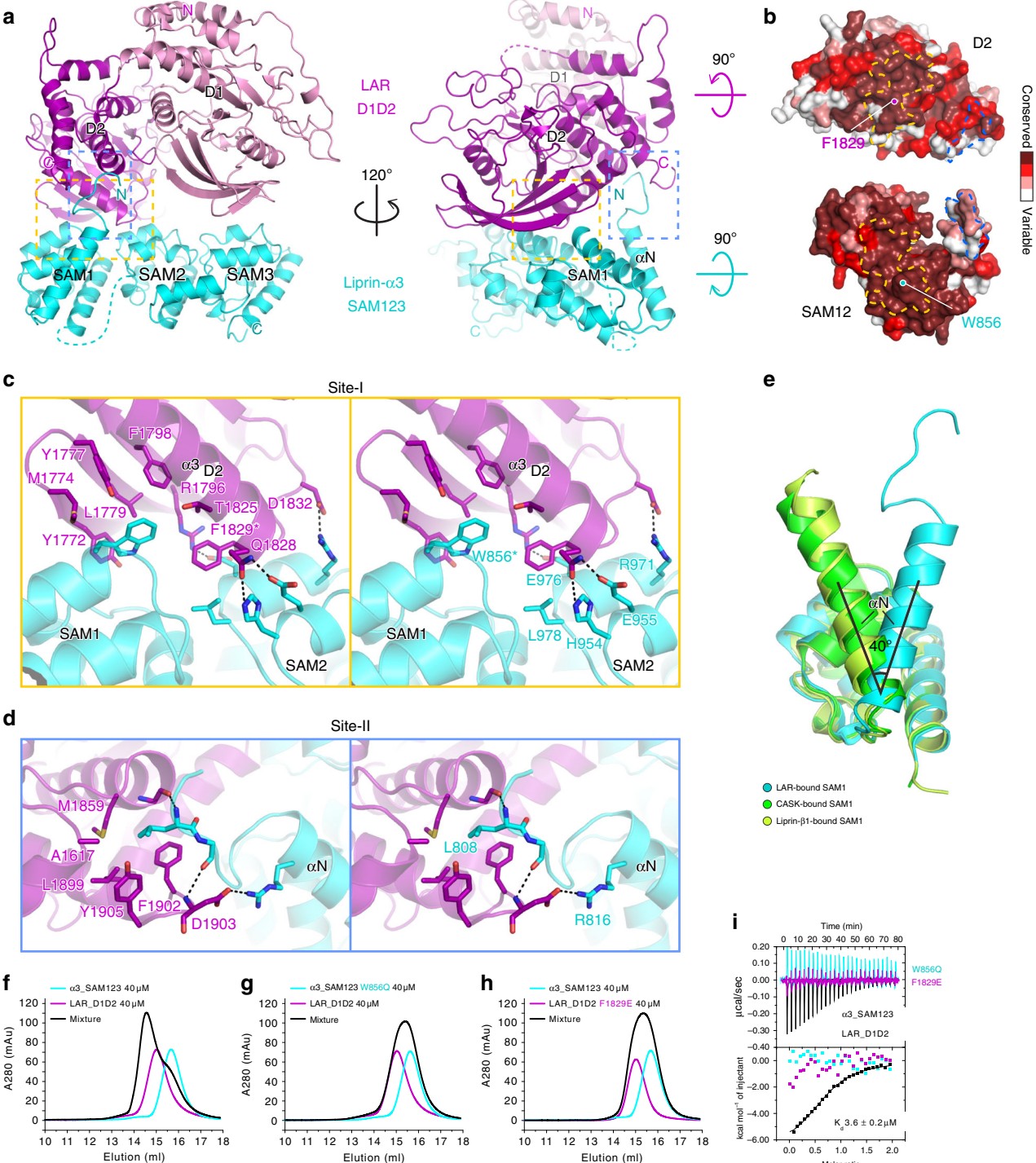

**Fig. 2 Structural characterization of the liprin-α3/LAR interaction. a** Ribbon representations of the α3_SAM123/LAR-D1D2 complex structure. The two binding sites were highlighted by dashed boxes. **b** Surface representations showing the high conservation of the two binding sites. **c, d** Stereoview of the atomic details of the two binding sites between α3_SAM123 and LAR-D1D2, corresponding to the boxed regions shown in (**a**) with the same color. W856 in liprin-α3 and F1829 in LAR, indicated by asterisks, play the key role in the liprin-α3/LAR interaction. Hydrogen bonds and salt bridges are indicated by dashed lines. **e** Structural alignment of the SAM1 domains from the complex structures of α3_SAM123/LAR-D1D2, α2_SAM123/CASK_CaMK (PDB ID: 3TAC), and α2_SAM123/ β1_SAM123 (PDB ID: 3TAD) showing the orientation change of the αN-helix. **f–h** Analytical gel filtration analysis showing that either the W856Q mutation in liprin-α3 or the F1829 mutation in LAR disrupts the liprin-α3/LAR interaction. **i** ITC-based measurement of the binding of α3_SAM123 or its W856Q mutant to LAR_D1D2 and the binding of α3_SAM123 to the F1829E mutant of LAR_D1D2.

C-terminal residues of the D2 domain through hydrophobic interactions, hydrogen bonds, as well as charge–charge interactions (Fig. 2a, d and Supplementary Figs. 1 and 2), supported by the weakened interaction between α2_SAM123ΔN and LAR_D1D2 (Table 1). Interestingly, compared to those in the CASK-bound and liprin-β1-bound structures of α2_SAM123 (Supplementary Fig. 5c), the αN-helix in the LAR-bound structure of α3_SAM123 shows a rotation of ~40° (Fig. 2e). Considering the limited interaction between αN and the rest part of the SAM1 domain, αN is likely to be dragged by the binding of

liprin-α to LAR. Nevertheless, we could not rule out the possibility that the change of the αN orientation is caused by the sequence variation between liprin-α2 and α3 (Supplementary Fig. 1). Although being close to the SAM3 domain (Fig. 2a), the D1 domain of LAR is not directly involved in the SAM123/D1D2 interaction. Since the LAR_D1D2 protein showed a higher thermal stability than the D2 domain only (Supplementary Fig. 5d), the D1 domain contributes to the binding of liprin-α to LAR probably by stabilizing the folding of the D2 domain through the D1/D2 interdomain interaction[15].

To validate our structural model, we designed two mutations by replacing the central interface residues of site-I, W856 and F1829 in α3_SAM123 and LAR_D1D2, respectively (Fig. 2b), to hydrophilic residues, like glutamine or glutamate, which presumably disrupt the hydrophobic interactions in site-I. Consistently, both of the W856Q and F1829E mutations abolished the α3_SAM123/LAR_D1D2 interaction (Fig. 2f–i). In addition, the F1829E mutant of LAR lost its binding to either liprin-α1 or α2 (Supplementary Fig. 5e, f). As several liprin-α proteins were used in this study, to simplify description, we hereafter used "WQ" to represent the corresponding mutations of W856Q in liprin-αs. Correspondingly, "FE" was used to represent the F1829E mutation in LAR.

**Liprin-α promotes cluster formation of LAR in cells**. To explore the functional implication of our structural findings, we over-expressed the full-length proteins of liprin-α1 and LAR in COS7 cells, as these two proteins have been reported to regulate cell motility in different cell lines[29,41–43]. The results showed that both liprin-α1 and LAR largely diffused in cells, although small puncta of LAR were occasionally observed (Fig. 3a, b). Interestingly, by co-expressing the two proteins, a huge amount of large puncta were formed in cells, which were usually clustered together to occupy the majority of the cell surface (Fig. 3c, h). In these large puncta, liprin-α1 and LAR were co-clustered (Fig. 3c). However, the co-clustering of liprin-α1 and LAR was eliminated by either the WQ mutation in liprin-α1 or the FE mutation in LAR (Fig. 3d, e, h), indicating that the SAM123/D1D2 interaction is required for the large cluster formation. Similarly, the strong co-clustering of liprin-α2 and LAR was also observed in COS7 cells (Supplementary Fig. 6).

In transfected cells, the majority of LAR was distributed at dorsal plasma membranes while liprin-α1 alone diffused in the cytoplasm (Supplementary Fig. 7a, b). Interestingly, the co-expression of liprin-α1 and LAR resulted in the redistribution of the two proteins to ventral plasma membranes (Supplementary Fig. 7c). In co-transfected cells, the ventral membrane localization of LAR was disrupted by introducing the WQ mutation to liprin-α1 (Supplementary Fig. 7d). The total internal reflection fluorescence (TIRF) microscopic data further confirmed that the co-clustering of liprin-α1 and LAR indeed happens at ventral plasma membrane regions (Supplementary Fig. 8). Together, the above cell imaging analysis suggests that the liprin-α regulate the cellular localization of LAR by facilitating cluster formation.

As the mixture of the purified SAM123 and D1D2 fragments did not form large aggregates (Fig. 1b), we suspected that the other regions of liprin-α are required for promoting the clustering of LAR. Considering that the N-terminal coiled-coils of liprin-α form oligomers[44], liprin-α may cross-link small clusters of LAR by self-oligomerization to form large clusters. Indeed, in cells co-expressing LAR and liprin-α1_SAM123 that does not contain the coiled-coils, the co-clustering was dramatically decreased (Fig. 3h), despite that liprin-α1 still co-localized with LAR in small puncta (Fig. 3f). In addition, co-expression of the cytoplasmic D1D2 region of LAR with liprin-α1 failed to generate

large puncta (Fig. 3g, h), implicating that the plasma membrane localization and/or the extracellular attachment may be required for the liprin-α-promoted LAR clustering.

**The clustering of LAR is mediated via the D1/D1 interaction**. Since the complex structure that we solved contains four LAR_D1D2 protomers, we analyzed the packing interfaces between them. As shown in Fig. 4a, the four protomers are packed side by side via the intermolecular interaction between two neighboring D1 domains, which were not found beyond the asymmetric unit. The three D1/D1 interfaces, each covering approximately 1000 Å$^2$ surface area, are highly similar with pre-dominantly hydrophilic interactions mediated mainly by the conserved residues in the loop regions of the D1 domain (Fig. 4b and Supplementary Figs. 2 and 9a), indicating that the D1/D1 packing in the crystal is unlikely to be a nonspecific interaction. Regarding our observation and other observations of the LAR-RPTP clustering[19,22,35], the side-by-side packing of the D1 domains allows the LAR-RPTP molecules to be assembled together to form large clusters (Supplementary Fig. 9b).

To confirm the involvement of the D1 domain in the cluster formation of LAR, we designed a RQN mutation in LAR, by replacing three hydrophilic residues, R1415, Q1417, and N1418, at the D1/D1 interface to alanine, which presumably disrupts the hydrophilic interactions between the D1 domains (Fig. 4b). If the cluster formation were mediated via the D1/D1 association, this RQN mutation would cause an impaired clustering of LAR. Consistent with our hypothesis, the liprin-α1/LAR co-clustering was largely diminished by the RQN mutation (Figs. 4c and 3h and Supplementary Fig. 10). In addition, liprin-α1 remained co-localized with the RQN mutant of LAR in small puncta (Fig. 4c), indicating that the RQN mutation interferes with the LAR clustering without affecting the liprin-α1/LAR interaction.

As the hydrophilic D1/D1 interface are largely solvent-exposed (Fig. 4b), the self-association of LAR_D1D2 was unstable and thereby not detectable in solution (Fig. 1b). To probe such a weak D1/D1 interaction, we used a site-specific chemical cross-linking approach. If the LAR_D1D2 molecules in solution interacts with each other through the same D1/D1 interaction observed in crystal, the substitution of both L1539 and T1593 that are close to each other in the D1/D1 interface (Fig. 4b and Supplementary Fig. 9a), with cysteine would specifically promote the self-association of LAR_D1D2 by forming intermolecular disulfide bridges. Indeed, the double-cysteine mutant of LAR_D1D2 formed reversible oligomers by removing reducing reagents in buffer, whereas neither the L1539C nor T1593C mutant showed an obvious oligomer formation (Fig. 4d, e).

**The clustering of LAR attenuates the LAR phosphatase activity**. Since the phosphatase activity of LAR is mainly mediated by the D1 domain[15], we asked whether the D1/D1 interaction has the potential to influence the catalytic function of the D1 domain. We found that the catalytic residue, C1548, sitting at the bottom of the substrate-binding pocket, is very close to the D1/D1 interface (Fig. 5a). The D1/D1 association positions a loop connecting the α3$_{D1}$-helix and the β11$_{D1}$-sheet in one D1 domain (D1$^A$) above the substrate-binding pocket of the other neighboring D1 domain (D1$^B$) (Fig. 5a, b). Several basic residues, surrounding the catalytic C1548 in this pocket, create a highly positively charged environment for recognizing phosphorylated substrates (Fig. 5b and Supplementary Fig. 11). Surprisingly, D1540, located at the tip of the α3$_{D1}$/β11$_{D1}$-loop of D1$^A$, inserts its negatively charged side-chain into the positively charged pocket of D1$^B$ (Fig. 5b). Despite that the distance between D1540 and the basic residues at the pocket is not close enough for forming strong salt bridges, the

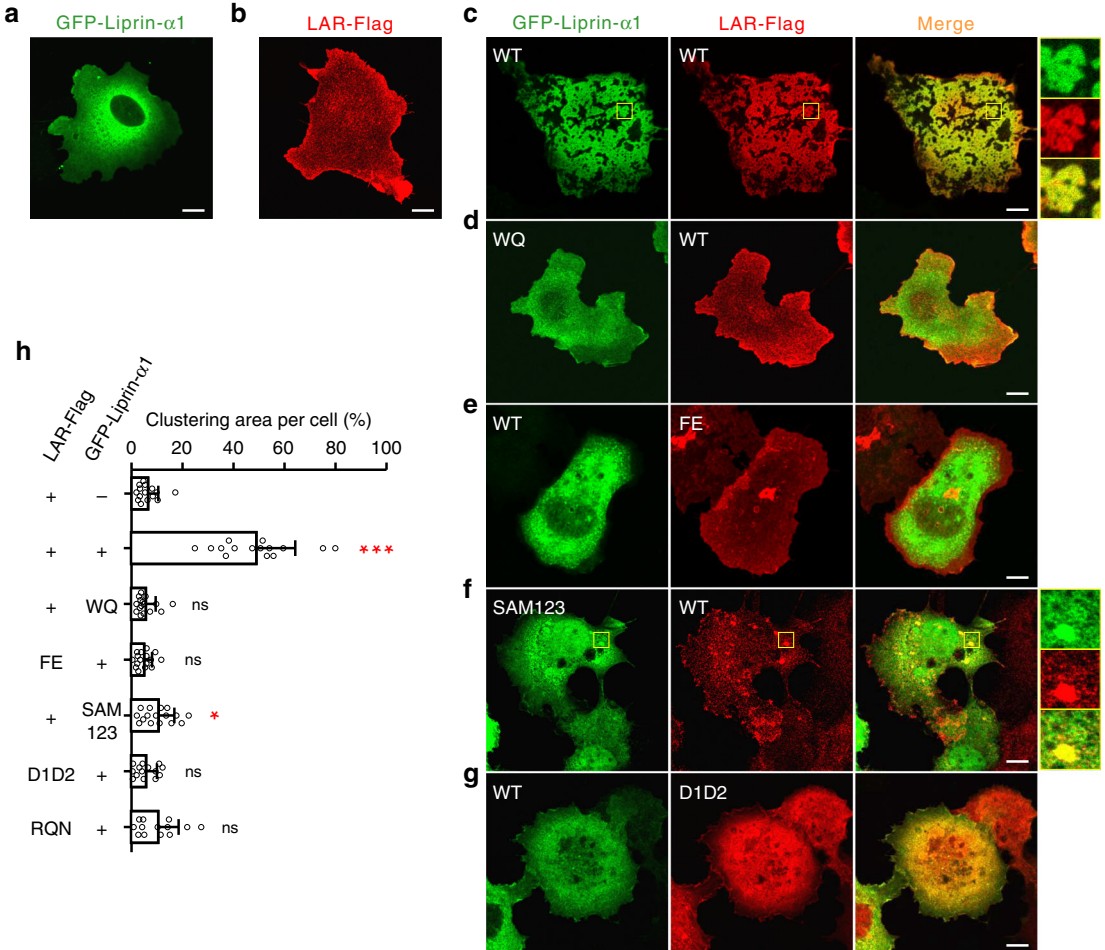

**Fig. 3 Liprin-α promotes the clustering of LAR. a, b** Cell imaging of over-expressed N-terminal GFP-tagged liprin-α1 (**a**) and C-terminal Flag-tagged LAR (**b**) in COS7 cells. **c–g** Cells co-transfected with GFP-tagged liprin-α1 wild-type or its variants and Flag-tagged LAR or its variants. Regions of interesting were highlighted by yellow boxes, enlarged four times, and aligned on the right of each merged image. "WQ" and "FE" indicate the W896Q mutant of liprin-α1 and the F1829E mutant of LAR, respectively. Scale bar, 10 μm. **h** Quantification of the LAR clustering levels as shown in (**a–g**) and in Fig. 4c. Error bars represent s.d. ~15 cells per experimental condition. The unpaired Student's $t$ test analysis was used to define a statistically significant difference (*$p <$ 0.05; **$p <$ 0.01; ***$p <$ 0.001; ns not significant).

positioning of a negatively charged residue near the substrate-binding pocket is very likely to block the entry of negatively charged substrates by both charge repulsion and steric effect, thereby inhibiting the enzymatic activity of LAR. Since D1540 is highly conserved in all LAR-RPTPs across species (Supplementary Fig. 2), the regulation mechanism of the LAR phosphatase activity by the D1/D1 interaction found here is likely to be shared by other LAR-RPTPs.

To measure the potential activity change of LAR mediated by the D1/D1 interaction, we performed in vitro phosphatase activity assay[45] using the double-cysteine mutant (L1539C/T1593C) of LAR_D1D2 in the monomeric and dimeric states, respectively. Based on our structural finding, one of the two active sites in the D1-mediated dimer is blocked while the other remains accessible for reaction (Fig. 5a). Consistently, the disulfide-linked LAR_D1D2 dimer had ~40% phosphatase activity of the monomeric mutant (Fig. 5c). In contrast, when adding 2 mM DTT in the reaction buffer, the phosphatase activity of the LAR_D1D2 dimer was largely recovered to approximately 80% compared to the monomer fraction (Fig. 5d), presumably due to the dissociation of the disulfide-linked LAR_D1D2 dimer in the reducing environment (Fig. 4d).

Next, since the D1/D1 interaction mediates the cluster formation of LAR, we asked whether the LAR clustering could downregulate the phosphatase activity of LAR in cells. To compare the phosphatase activity of LAR with or without the liprin-α-promoted clustering, we probed phosphotyrosine (pY) levels in cells transfected with LAR alone or co-transfected with LAR and liprin-α1. By using anti-pY antibody, we detected a significant difference of the total pY levels between the cells expressing wild-type LAR and its catalytically inactive CS mutant, by replacing catalytic C1548 to serine in the D1 domain (Fig. 5e, f), confirming that the change of the LAR activity can be reflected by the change of the pY level. Consistent with our hypothesis, the phosphatase activity of LAR is inhibited by the LAR clustering, as indicated by the observation that the pY signal was increased to a relatively high level in the cells co-expressing LAR-Flag with GFP-liprin-α1 (Fig. 5e). Introducing the binding deficient mutation in either liprin-α1 (WQ) or LAR (FE) leads to the release of the inhibition, as indicated by the low pY level comparable to the cells expressing LAR-Flag alone (Fig. 5e). A similar decreased pY signal was observed in cells co-expressing the RQN mutation of LAR that disrupts the D1/D1 interaction (Fig. 5e). As the LAR clustering is disrupted by either the WQ,

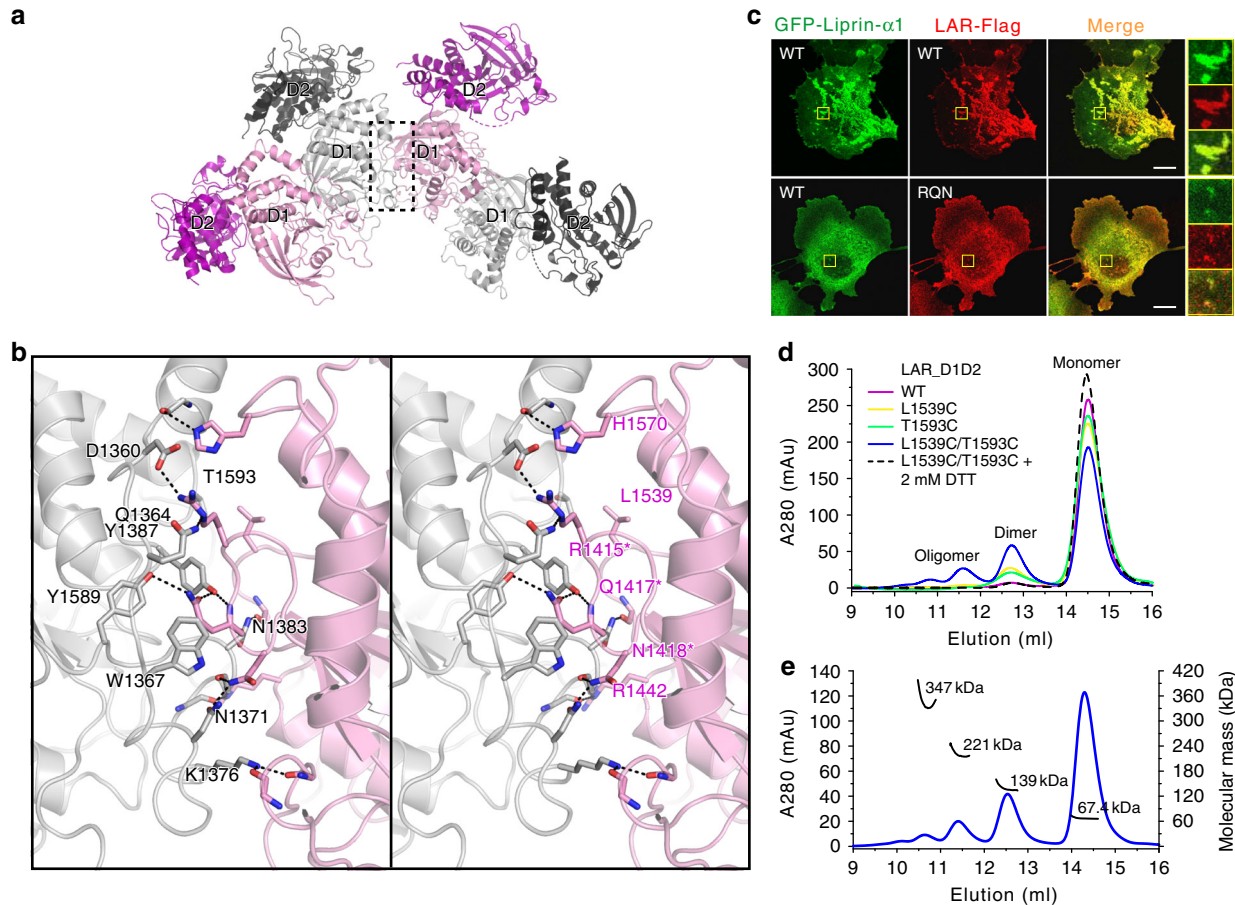

**Fig. 4 The D1/D1 packing mediates the clustering of LAR. a** The D1/D1 interaction found between the four LAR-D1D2 protomers in one asymmetric unit. **b** Stereoview of the atomic details for one LAR-D1D2 packing interface boxed in (**a**). The other two D1/D1 packing interfaces were shown in Supplementary Fig. 9a. Hydrogen bonds and salt bridges were indicated by dashed lines. Residues (R1415, Q1417, and N1418) mutated to alanine in the RQN mutation of LAR were indicated by asterisks in the right panel. **c** COS7 cells co-transfected with GFP-liprin-α1 and wild-type LAR-Flag or its RQN mutant. Regions of interest were highlighted by yellow boxes and enlarged four times on the right of each merged image. Scale bar, 10 μm. **d** Analytical gel filtration analysis showing the oligomer formation induced by the L1539C/T1593C double-cysteine mutation in LAR_D1D2. **e** Molecular weight measurement of different oligomeric fractions of the double-cysteine mutant by using multiangle static light scattering. The theoretical molecular weight of LAR_D1D2 is 65.9 kDa.

FE, or RQN mutation, the elevated phosphatase activity is very likely to be caused by the low clustering level of LAR. In contrast, when expressing the CS mutant in cells, co-expressing either the wild-type protein or the WQ mutant of liprin-α1 affected little on the high pY level (Fig. 5e, f), ruling out the possibility that liprin-α1 regulates the pY level through a LAR-independent manner.

**Liprin-α binding to LAR and liprin-β is mutually exclusive.** Previously, we determined the structures of liprin-α2_SAM123 in complex with CASK and liprin-β1 and found that α2_SAM123 is capable of interacting with CASK and liprin-β simultaneously[32]. After solving the α3_SAM123/LAR_D1D2 complex structure, we compared the target-binding surfaces on liprin-α_SAM123 by superimposing the three liprin-α_SAM123 structures in these three complexes. Apparently, the LAR-binding surfaces on liprin-α_SAM123 do not overlap with the other two binding surfaces for CASK and liprin-β (Fig. 6a), suggesting that the binding of LAR to liprin-α would not interfere with the binding of CASK and liprin-β. Indeed, when adding CASK_CaMK to the α2_SAM123/LAR_D1D2 complex, we detected a larger complex formation in gel filtration column (Supplementary Fig. 12a), supporting that liprin-α can interact with LAR and CASK simultaneously.

However, to our surprise, the presence of LAR_D1D2 prevents α2_SAM123 from binding to β1_SAM123 (Fig. 6b–d).

Since the SAM1 domain participates in the binding to both of LAR and liprin-β, we carefully dissected structural changes in this domain. Despite the obvious change of the αN orientation (Fig. 2e), no structural evidence suggests the involvement of the αN-helix in the liprin-α/liprin-β interaction. Interestingly, we found that the indole ring of W856 in liprin-α3, the residue critical for the liprin-α/LAR interaction (Fig. 2g, i), rotates 180° upon the LAR's binding (Fig. 6e). This conformational change of W856 allows an adjacent residue, Y857 flips its sidechain from the position pointing to the hydrophobic core of the SAM1 domain to the position pointing to the outside. The sidechain flipping of Y857 leaves a space for I883 to further plug its hydrophobic sidechain deeper inside, resulting in a 3 Å-shift of the loop where I883 locates. The loop-shift drags E882, another charged residue in this loop, downwards to cover the liprin-β-binding pocket in the SAM1 domain. Therefore, the initial subtle change of W856 induced by the binding of LAR to liprin-α3 propagates to the liprin-β-binding site and thereby allosterically inhibits the binding of liprin-α3 to liprin-β1, not only by generating steric hindrance but also by disturbing the hydrophobic interactions found in the liprin-α/liprin-β interaction[32] (Fig. 6e).

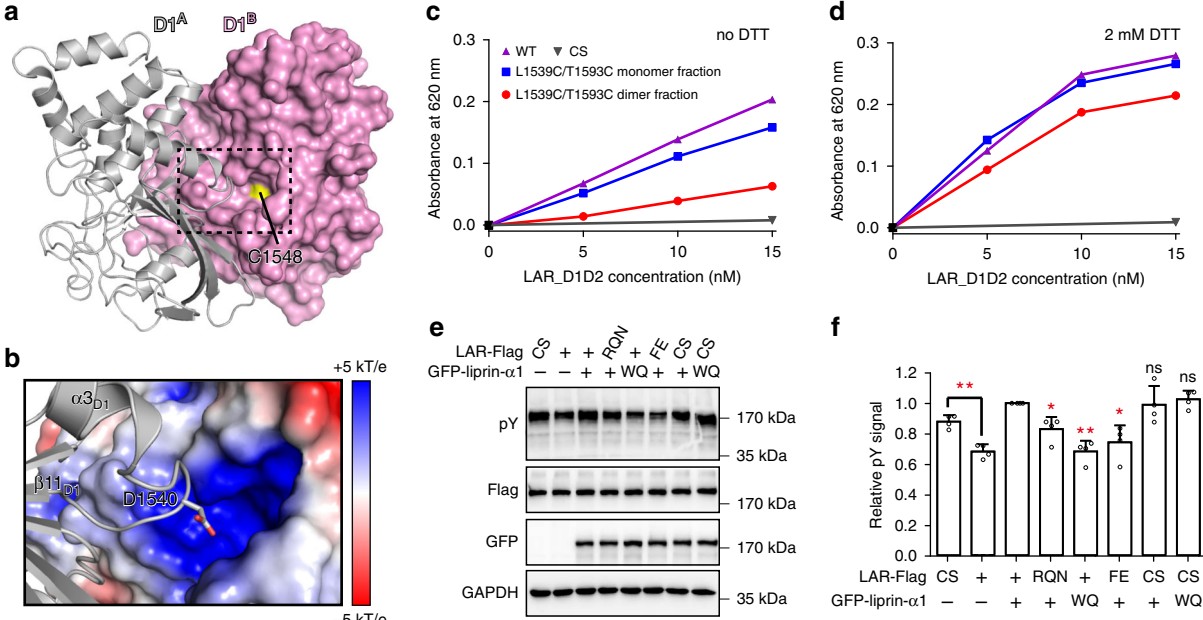

**Fig. 5 The D1/D1 interaction interferes with the phosphatase activity of LAR. a** Structural analysis of the steric blocking of the substrate entry by the D1/D1 interaction. The two neighboring D1 domains, named D1$^A$ and D1$^B$, were shown as white ribbons and magenta surface, respectively. The substrate-binding pocket was boxed. The catalytic residue, C1548 was highlighted in yellow. **b** Surface charge distribution of the substrate-binding pocket. D1540 in the α3$_{D1}$/β11$_{D1}$-loop was shown as stick mode. **c, d** Phosphatase activity analysis of LAR_D1D2 and its double-cysteine mutant in a reaction buffer without (**c**) or with DTT (**d**). The catalytically inactive C1548S (CS) mutant was used as the negative control. **e** Detection of total phosphorylated tyrosine (pY) in cells transfected with LAR and its CS mutant alone, or co-transfected with various variants of LAR and liprin-α1. Immunoblots were probed with indicated primary antibodies. **f** Quantification of pY intensities as shown in (**c**). Data were obtained from four independent experiments. Intensities were normalized by setting the pY intensity measured from cells expressing wild-type LAR-Flag and GFP-liprin-α1 as 1. Error bars represent s.d. The paired Student's *t* test analysis was used to define a statistically significant difference (*$p < 0.05$; **$p < 0.01$; ***$p < 0.001$; ns not significant).

As liprin-α1, liprin-β1, and LAR have been reported to play roles in integrin-mediated focal adhesion (FA)[29,41–43,46,47], we analyzed the potential effects of the competitive binding of liprin-β and LAR to liprin-α on the cellular localization of liprin-β. In COS7 cells, the endogenous proteins of liprin-α1 and liprin-β1 were co-localized at the peripheral regions of the FA, an integrin-mediated cell–ECM adhesion site[48], using paxillin as the FA marker (Fig. 6f). However, in the LAR-transfected cells, the majority of liprin-β1 shifted from the peripheral regions to the central regions of the FAs (Fig. 6g and Supplementary Fig. 12b), suggesting that the overexpressed LAR protein disrupts the liprin-α1/liprin-β1 interaction and causes the mislocalization of liprin-β1. Consistently, the liprin-α-binding deficient mutant of LAR (FE) did not interfere with the FA peripheral localization of liprin-β1 (Fig. 6h and Supplementary Fig. 12b).

## Discussion

Clustering of cell surface receptors has been widely found to regulate numerous cellular activities, including cell adhesion, migration, and synaptic signaling[49–51]. It is intriguing to explore receptor clustering in coordinating extracellular binding and intracellular signaling. Previous studies focused on the extracellular interaction induced clustering of LAR-RPTPs[19–21]. In this study, we found that the intracellular binding of liprin-α to LAR also plays a vital role in the cluster formation of the receptor (Fig. 3). We argued that the binding of LAR-RPTPs to their extracellular and intracellular binding partners is coordinated in generating the proper clusters of receptors. In agreement with this hypothesis, both extracellular Slitrk1 and intracellular liprin-α2 and -α3 were found to regulate PTPσ-mediated presynaptic assembly[31]. These regulatory proteins promote the cluster

formation of LAR by positioning the receptors spatially close to each other[19,20] (Fig. 4a) and thereby enhancing the weak D1-mediated self-association. We note with interest that recent structural studies of deleted in colorectal cancer (DCC), a netrin receptor, important for axon navigation, suggests the associations of DCC with both its extracellular and intracellular binding partners play roles in mediating the receptor clustering[52–55].

The phosphatase activity regulation mechanism of LAR-RPTPs has been largely a mystery, despite that previous studies on other types of RPTPs suggested an inhibition mechanism, in which the catalytic D1 domain forms a homodimer via a head-to-head manner[56–58] (Supplementary Fig. 9b). However, no similar dimer of the LAR-RPTP D1 domain was observed in this or other studies[15]. Instead, our structure unveils a side-by-side association mode of the D1 domain (Supplementary Fig. 9a) that not only mediates the cluster formation of LAR (Fig. 4), but also prevents substrates from entering the active site (Fig. 5). Notably, the D1/D1 packing interface is mainly mediated by hydrophilic interactions (Fig. 4b). Since the interface is largely solvent-exposed, the self-association between the LAR_D1D2 molecules is unlikely to be detected in solution, which explain the reason why the similar D1/D1 packing was not observed in the previous structural study of apo LAR_D1D2[15] and why the oligomerized liprin-α is required for the massive clustering of LAR (Fig. 3f).

In addition to promote the clustering, liprin-α also regulates the cellular localization of LAR[22,35]. The LAR clusters were primarily localized to the ventral plasma membrane when co-expressed with liprin-α (Supplementary Figs. 7 and 8). In adherent cultured cells, numerous cell–ECM adhesion sites (e.g., FA) locate at the ventral plasma membrane to connect cells with the ECM[59]. Considering that LAR-RPTPs are actively involved in the cell-ECM interactions, the ventral plasma membrane

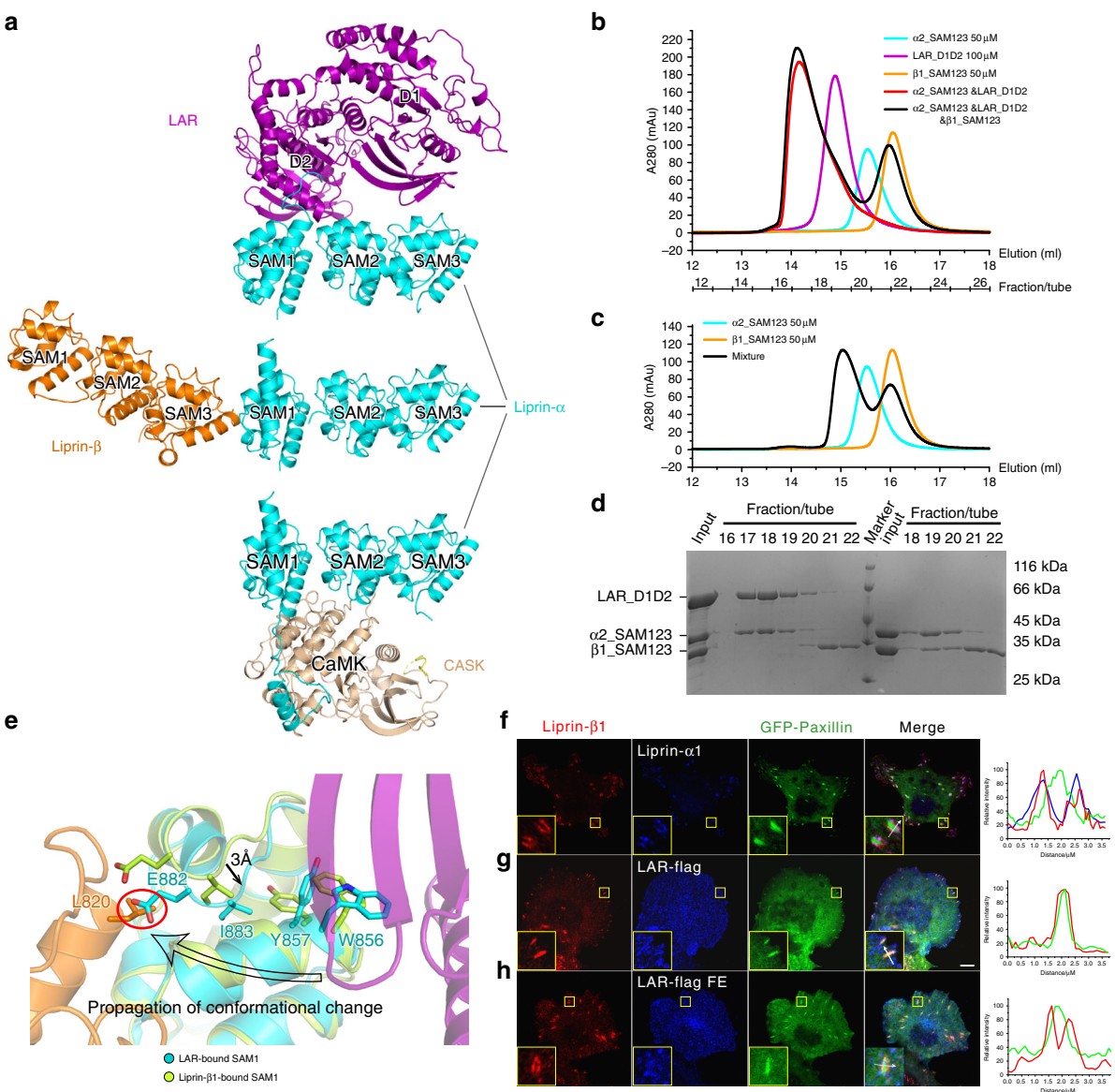

**Fig. 6 Allosteric regulation of SAM123-mediated interactions. a** Analysis of the different target-binding surfaces on liprin-α_SAM123. The three liprin-α complex structures are presented with liprin-α_SAM123 in the same orientation. **b, c** Analytical gel filtration analysis showing the binding of liprin-α2_SAM123 to LAR with or without liprin-β1 (**b**) and to liprin-β1 (**c**). **d** SDS-PAGE analysis of the collected gel filtration fractions of the α2_SAM123/LAR_D1D2/β1-α_SAM123 mixture and the α2_SAM123/β1-α_SAM123 mixture as shown in (**b**) and (**c**), respectively. **e** Structural comparison of the LAR-bound SAM1 and liprin-β1-bound SAM1. Residues with large conformational or positional changes were shown in stick model. The steric crash between E882 in the LAR-bound SAM1 and L820 in liprin-β1, the critical liprin-α-binding site, was highlighted by a red circle. **f, g** Cell imaging analysis of cellular localization of liprin-β1. Endogenous liprin-α1 and liprin-β1 co-localized in the peripheral regions of the focal adhesion (FA), marked by GFP-paxillin (**f**). The localization of endogenous liprin-β1 was further analyzed in cells transfected with wild-type LAR-Flag (**g**) and its F1829E mutant (**h**). Scale bar, 10 µm. Regions of interesting were highlighted and enlarged as insets. Results were further confirmed by line profile analysis shown on the right.

localization driven by liprin-α-promoted cluster may efficiently enhance the connections between LAR-RPTPs and their ECM ligands at the adhesion sites and in turn further promote the cluster formation of LAR.

The fascinating capability of liprin-α in organizing various protein-protein interaction via the three SAM domains provides the assembly mechanism of target proteins for different cellular activities[32]. Our finding that LAR and liprin-β allosterically compete with each other for the binding to liprin-α (Fig. 6) advances the understating of the liprin-α-mediated complex formation. In addition, the binding of LAR to liprin-α induces a large rotational change of the αN-helix (Fig. 2e). Given the high

involvement of the SAM1 domain in target binding, we speculate that the αN rotation may create or disrupt a binding surface for an unknown binding partner of liprin-α.

In summary, we demonstrated that liprin-α, through its interaction with the cytoplasmic D1D2 domain of LAR, promotes the cluster formation and ventral plasma membrane localization of LAR in cells. Importantly, the LAR clustering requires the homophilic interaction mediated by the catalytically active D1 domain. The D1/D1 interaction interferes with the substrate-binding pocket at the D1 domain and thereby attenuating the phosphatase activity of LAR. This intracellular control of the LAR clustering unveils a regulatory mechanism of the activities of the

receptor-type phosphatase and provides insights into the understanding of the cluster formation for cell surface receptors.

## Methods

**Expression constructs and site-directed mutagenesis.** DNA encoding sequences of human liprin-α2_SAM1, human liprin-α2_SAM23, human liprin-α2_SAM123, human liprin-α1_SAM123, mouse liprin-α3_SAM123, mouse liprin-α4_SAM123, mouse liprin-β1_SAM123, human PTP-σ_D1D2, and human PTP-δ_D1D2 were subcloned into a modified pET-32a vector with an N-terminal thioredoxin (Trx)-His$_6$-tag. Human LAR_D2 and LAR_D1D2 were subcloned into a modified pET-28a vector with an N-terminal His$_6$-SUMO-tag. The liprin-α1, liprin-α2, and paxillin constructs for cellular assays were subcloned into a mammalian-expression vector containing an N-terminal GFP tag, while LAR were cloned into a mammalian-expression vector with a C-terminal Flag tag. The constructs and primers used in this study were summarized in Supplementary Tables 1 and 2, respectively. All point mutations were created using site-directed mutagenesis kit and confirmed by DNA sequencing.

**Protein expression and purification.** All of the proteins were expressed in Rosetta BL21(DE3) *E. coli* cells with 0.2 mM IPTG induced in LB medium at 16 °C. Cells were resuspended in the binding buffer containing 50 mM Tris–HCl pH 7.5, 500 mM NaCl, 5 mM imidazole, and 1 mM phenylmethylsulfonyl fluoride and lysed using ultrahigh-pressure homogenizer (ATS, AH-BASICI). The lysate was spun at 48,384 *g* for 30 min. The supernatant was loaded directly onto a Ni-NTA agarose column (Qiagen) that had been equilibrated with the binding buffer. The target protein was eluted with buffer containing 50 mM Tris–HCl pH 7.5, 500 mM NaCl, and 500 mM imidazole. The eluted proteins were loaded onto a HighLoad 26/60 Superdex-200 size-exclusion column (GE Healthcare) and were then eluted with buffer containing 50 mM Tris-HCl pH 7.5, 100 mM NaCl, 1 mM EDTA, and 1 mM DTT. For isothermal titration calorimetry and analytical gel filtration chromatography, the protein samples contained the N-terminal tags. For crystallization, the N-terminal Trx-His$_6$-tag or His$_6$-SUMO-tag was cleaved by HRV 3C protease and SUMO protease, respectively, and removed by size-exclusion chromatography.

**Crystallization and data collection.** The complex samples were prepared by mixing liprin-α3_SAM123 with LAR_D1D2 and were further purified by size-exclusion chromatography with buffer containing 50 mM Tris-HCl pH 7.5, 200 mM NaCl, 1 mM EDTA, and 1 mM DTT. The complexes were concentrated to ~20 mg/ml for crystallization. Crystals were obtained by the sitting drop vapor diffusion method at 16 °C. To set up a sitting drop, 1 μl of concentrated protein solution was mixed with 1 μl of crystallization solution with 0.2 M Tris-HCl (pH 8.0), 12% w/v PEG20,000 and 0.1 M Adenosine-5′-triphosphate disodium salt hydrate. Before X-ray diffraction experiments, crystals were soaked in the crystallization solutions containing additional 30% w/v glycerol for cryoprotection. Diffraction data were collected at the Shanghai Synchrotron Radiation Facility beamline BL17U1, BL18U1, and BL19U1. Data were processed and scaled using HKL3000 software.

**Structure determination and analysis.** The initial phase of the complex structure was determined by molecular replacement in PHASER using the apo structures of LAR_D1D2 (PDB ID: 1LAR) and liprin-α2_SAM123 (PDB ID: 3TAC) as the search models. The model was refined in PHENIX[60]. COOT[61] was used for model rebuilding and adjustments. In the final stage, an additional TLS refinement was performed in PHENIX. The model quality was check by MolProbity[62]. The final refinement statistics are listed in Table 2. All structure figures were prepared by using PyMOL (https://www.pymol.org).

**Isothermal titration calorimetry (ITC) analysis.** ITC experiments were carried out on a VP-ITC Microcal calorimeter (Malvern) at 25 °C. All proteins were dissolved in buffer containing 50 mM Tris-HCl pH 7.5, 100 mM NaCl, 1 mM EDTA, and 1 mM DTT. Each titration point typically is consisted of injecting 10 μl aliquots of the liprin-α fragments or their mutants, at concentration of 200 μM into the solution of containing LAR_D1D2, its mutants, LAR_D2, PTP-σ_D1D2, or PTP-δ_D1D2 at a concentration of 20 μM. A time interval of 150 or 180 s between two titration points was used to ensure the complete equilibrium of each titration reaction. The titration data were analyzed using the program Origin7.0 and fitted by a one-site binding model.

**Analytical gel filtration chromatography.** Analytical gel filtration chromatography was carried out on an ÄKTA pure system (GE Healthcare). For disulfide-bond induced oligomer formation, LAR_D1D2 and its cysteine mutants were purified and then incubated for 3 days at room temperature in a buffer containing 50 mM Tris-HCl pH 7.5, 100 mM NaCl, and 1 mM EDTA. Protein samples at indicated concentrations were loaded onto a Superdex 200 Increase 10/300 GL column (GE Healthcare), equilibrated with a buffer containing 50 mM Tris-HCl pH 7.5, 100 mM NaCl, 1 mM EDTA. DTT was fleshly added into the buffer with a final concentration of 1 mM or indicated otherwise.

**ThermoFluor assay.** Protein thermal stability was measured by using a fluorescent protein-binding dye SYPROÒ Orange (Sigma-Aldrich, S5692). The experiments are conducted utilizing the melt capability of a real-time PCR system (Applied Biosystems, ABI StepOne Plus qPCR). Protein concentration was 1 mg/ml with a buffer containing 50 mM Tris-HCl pH 7.5, 100 mM NaCl, 1 mM EDTA, and 1 mM DTT.

**Multiangle static light scattering.** A miniDAWN TREOS (Wyatt Technology Corporation) coupled with an ÄKTA pure system (GE Healthcare) was used for molar mass measurement. The procedure followed the protocol used for analytical gel filtration analysis.

**Protein tyrosine phosphatase assay.** Tyrosine phosphatase assay was performed following the instruction of the protein tyrosine phosphatase assays kit (Sigma-Aldrich, PTP-101). A monophosphate peptide of insulin receptor (TRDIY(p) ETDYYRK) was used as substrate for the LAR-catalyzed reaction. The reaction took 10 min at room temperature. Malachite Green phosphate assay kit (Cayman, 10009325) was used to determine the free phosphate generated by the dephosphorylation reaction. Experiments were repeated three times.

**Cell culture, transfection, and fluorescence imaging.** COS7 cells (Shanghai Cell Bank, Chinese Academy of Sciences) or HEK293T (ATCC) cells were cultured in Dulbecco's modification of Eagle's medium (DMEM) (Coring, 10-013-CVR) supplemented with 10% fetal bovine serum (Pan Biotech) and 50 U/ml penicillin and streptomycin. Transfections of indicated plasmids were performed with PEI (polyethylenimine-25,000, Polyscience) according to the manufacturer's instructions. One day after transfection, the cells were trypsinized, replated on ~20 μg/ml fibronectin (Millipore)-coated coverslips and cultured for 1 h. After fixation with 4% paraformaldehyde, the cells were stained with indicated primary antibodies followed by fluorescent dye-conjugated secondary antibodies, and observed under a Nikon A1R confocal microscope or Nikon super-resolution microscope for TIRF microscopy. For immuno-fluorescence, primary antibodies against Flag (Sigma, F1804, dilution 1:200), liprin-α1 (ProteinTech, 14175-1-AP, dilution 1:200 and Santa Cruz, SC376141, dilution 1:50), and liprin-β1 (ATLAS, HPA001924, dilution 1:200) were used. Dye-conjugated secondary antibodies against rabbit IgG (H + L) (Thermo, A-11012) and mouse IgG (H + L) (Thermo, A-21203&A-21046) were diluted 1:500. Images were analyzed using Image J software (NIH, USA).

**Cell lysis and immunoblotting.** One day after transfection, HEK293T cells were collected and resuspended by a lysis buffer containing 50 mM Tris-HCl pH 7.5, 100 mM NaCl, 10 mM EDTA, 1× protease inhibitor cocktail (TargetMol, C001), 1× phosphatase inhibitor cocktail (MedChem Express, HY-K002 & HY-K0023), 1% Triton X-100. Sodium dodecyl sulfate (SDS) loading buffer was immediately added to the cell lysate. Samples were boiled for 15 min, separated by SDS-polyacrylamide gel electrophoresis (PAGE), and transferred to polyvinylidene difluoride (PVDF) membrane (Millipore). The membranes were subsequently blocked with 3% BSA in TBST (50 mM Tris-HCl, pH 7.4, 150 mM NaCl, and 0.1% Tween 20) for 1 h. The PVDF membranes were immunoblotted with antibodies against pTyr(PY99) (Santa Cruz, SC7020, dilution 1:100), GFP (TransGen, HT801, dilution 1:3000), Flag (Sigma, F1804, dilution 1:3000) or GAPDH (TransGen, HC301, dilution 1:3000) at room temperature for 1 h, and then probed with horseradish-peroxidase-conjugated secondary antibodies with a dilution of 1:10,000 (CellSignaling, 7076s) and developed with a chemiluminescent substrate (Millipore). Protein bands were visualized on the Tanon-6011C Chemiluminescent Imaging System (Tanon Science and Technology). All experiments were repeated at least four times.

**Reporting summary.** Further information on research design is available in the Nature Research Reporting Summary linked to this article.

## Data availability

The authors declare that all relevant data are available within the article and its Supplementary files or from the corresponding author upon reasonable request. The coordinates and structure factors for the liprin-α3_SAM123/LAR_D1D2 complex have been deposited in the Protein Data Bank (PDB) under the accession codes 6KR4 [https://doi.org/10.2210/pdb6KR4/pdb]. The source data underlying Figs. 3h, 5c–f, and Supplementary Figs. 6f, 8g, 10b, and 12b are provided as a Source Data file.

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

## Acknowledgements

We thank the assistance of Southern University of Science and Technology (SUSTech) Core Research Facilities. We thank the staff from BL17U, BL18U and BL19U1 beamlines of Shanghai Synchrotron Radiation Facility for assistance during data collection. This

work was supported by the National Natural Science Foundation of China (Grant nos. 31770791 and 31570741 to Z.W. and 31870757 to C.Y.), Natural Science Foundation of Guangdong Province (2016A030312016), Science and Technology Planning Project of Guangdong Province (2017B030301018), Shenzhen-Hong Kong Institute of Brain Science, Shenzhen Fundamental Research Institutions (2019SHIBS0002), and Shenzhen Science and Technology Innovation Commission (JCYJ20160229153100269 to Z.W. and JCYJ20160301112450474 to X.X.). Z.W. is a member of the Brain Research Center, SUSTech.

## Author contributions

Z.W. conceived and supervised the study. X.X., L.L., and M.L. designed the constructs, purified the proteins, and performed the biochemistry assays with help from W.Z., X.X., L.L., and Z.W. solved and analyzed the structures. X.X. performed the cellular assays. X.X., C.Y., and T.Z. analyzed the cellular data. X.X., C.Y., and Z.W. wrote the paper.

## Competing interests

The authors declare no competing interests.
