## [Peer Review File · Nature Communications]

Reviewers' comments:

Reviewer #1 (Remarks to the Author):

This manuscript by Xie et al presented high-resolution crystal structure of LAR D1D2/liprin-alpha3 SAM complex, revealing two conserved key sites involved in binding of LAR-RPTPs with liprin-alpha. Authors next performed cellular experiments to show that liprin-alpha facilitates LAR clustering by inducing oligomerization of liprin-alpha. Authors then found that LAR D1 mediates its homophilic interaction, which is critical for liprin-alpha-mediated LAR clustering and regulation of phosphatase activity. Authors lastly showed that LAR binding to liprin-alpha and liprin-beta is mutually exclusive using various methodologies.

Given the emerging importance of LAR-RPTPs in mediating various aspects of synapse formation and development, I think that this manuscript investigates timely and important topics. In particular, this manuscript (although performed in heterologous cells) addresses unknown roles of LAR-liprin-alpha interaction, which should be further evaluated in followup studies. Overall, data quality is very high and paper was well written. I support the publication of this manuscript in Nature Communications, should the following points are completely and sufficiently addressed:

Major Comments:

1. Quantification of clustering by liprin variants (WT and WQ) and/or LAR mutants (WT, FE, RFN, D1D2, etc) could strengthen the manuscript (Figs.3, 4 and 6).
2. Data in panel c of Fig.5 are qualitatively appropriate, but not suitable for quantification as presented in panel d in the same figure. Did the authors perform this experiment using ECL or I125-based autoradiography? ECL-based immunoblot data are not appropriate for making clear conclusions, given marginal difference in some of group comparisons.
3. Altered phosphatase activity of LAR induced by interference of D1/D1 interaction is very intriguing notion. Could authors directly test that liprin-alpha WQ mutant compromises the tyrosine phosphorylation level of a subset of LAR substrates (e.g. beta-catenin)?
4. What is the physiological significance of LAR binding to liprin-beta? To my knowledge, role of liprin-beta has not been seriously investigated, even in heterologous cells. Does liprin-beta similarly induce LAR clustering?

Minor Comments:

1. Typographical errors should be corrected: e.g. line 32, "involved" and line 267 "do not overlapped" are grammarly wrong.

Reviewer #2 (Remarks to the Author):

NCOMMS-19-28108, Xie et al.

Title: Structural Basis of Liprin-alpha-promoted LAR-RPTP Clustering for Modulation of Phosphatase Activity

The Authors report the complex structure of Liprin-alpha3 and the synaptic adhesion protein leukocyte common antigen-related receptor protein tyrosine phosphatase (LAR-RPTP) D1-D2 intracellular domains, and further more suggest a dimeric interaction between the D1 catalytic domains and suggest a model for clustering of the receptors.

The work is of high impact towards understanding the function of RPTPs as presynaptic organizers and role of Liprins in this, and provide lot of interesting new data. Data presented is sound, and text is well written. The paper is be suitable for publications in Nat. Communications with the clarifications provided as requested below:

1. Biochemical characterization – SAM 1 and SAM23 alone are shown not to show detectable binding – ITC data on this is lacking? Can you show that, or other direct binding measurement data, as the SEC data is not very quantitative.

Most importantly the SAM12 construct should be shown also, if this is the deduced functional fragment. What is the affinity of that for the D1D2 construct? I think this would be essential to have here.

2. Structure of the complex – the figures should be improved – green and red are not a good combination for colour-blind. I would change the colour-code.

Fig. 2 stereo figures (c and d) are too small to be seen in stereo (this seems to be overall the problem please enlarge so that they can be viewed (centers of figures should be ca. 6 cm away from each other), also Fig 2e is very small, and need to be larger to provide information.

It is mentioned that in the asymmetric unit there were four copies of LAR_D1D2 and three copies of SAM123 were “clearly traced”– can the authors explain this in more detail what is meant by this and explain the packing – This is shown in Suppl. Fig 4 but it is not explained that apparently one SAM123 construct is not visible in the maps. Any reasons for this? Also see below.

Regarding the binding interfaces – only selected residues are shown. Please report the various interface areas and provide a list of all the contacts e.g. in the supplement. (for both D1-D1 and SAM123 and Liprin interfaces).

Figure 4. a,b the colours are too pale to properly see, please make the images clearer. Same applies for figure 5, and the supplementary figures with the same grey-pale pink colouring.

Also please make the electron density figures in Suppl. Figure 4 larger for better visibility, So the trace can be actually seen. The size of the supplementary figures are not particularly restricted? It appears in the figure only SAM1 is shown not SAM123 with e-density although this is mentioned in the legend can you provide please the coordinated and maps for viewing of this? Based on the PDB validation report quite many residues are missing or the fit to density is not very good for the SAM-construct for last two chains. Size of the crystallized and used constructs should be mentioned (number of residues, MW for both proteins and missing residues / regions noted).

Minor points: helices and strands are simply referred to as alpha and beta without explanation e.g. line 13, for clarity please add “sheet” and “helix”, and throughout the text. (Again please provide a list of the most important interactions in the text and whole list for each interface in supplement, thank you).

Line 142 “hydrogen bondings” please check for language (hydrogen bonds). Line 144-146 CASK and Liprin beta interaction are already discussed here but presented in fig. 6. Could you refer to the figure here to make it clearer?

Stability of D1 and D1-D2 is mentioned, have you tested the stability of these constructs if that really is the cause for the difference in affinity? (e.g. by thermoflour assay), could this be done? Alternatively it maybe the four-fold difference measured is not significant in reality? Is this seen by other methods than ITC? Can you comment?

Line 162 and onwards the mutation code is little bit hard to follow so it would be easier if the numbers could be referred to somehow.

3. Cluster formation

Mutations nicely show that the D1-D1 interaction is required. It is not very clearly presented though as it is in two different places, and first its said soluble D1D2 failed to generate puncta? But how about in solution? Do you see large oligomers forming? (This is discussed in discussion perhaps state here if you don't see those and expand the discussion?)

Overall if possible make clearer the case that membrane attachment is needed, if that appears to be the case, could be in discussed more in discussion.

Also how does the oligomerization propagate beyond the asymmetric unit? Please mention this in the text somewhere.

Line 203 "side-by-side packing fashion" correct the language please.

Phosphotyrosine assay: is the real ligand for phosphorylation known? Any possibility to follow that?

Line 246 "enzyme-dead"  catalytically inactive.

Discussion: see for typos, L308 " bindings"

Please elaborate on the role of ventral plasma membrane localization here (explain why it is important observation, this is only hinted at, please spell it out clearly).

Please end with a conclusive statement about the current study, not a speculative one which is beyond the current data.

Point-by-point Response

Before a point-by-point response to the referees' comments, we thank both reviewers for recognizing the novelty and importance of our works and their critical and constructive suggestions that help us to efficiently improve the manuscript.

Our responses are shown in blue.

Reviewer #1 (Remarks to the Author):

This manuscript by Xie et al presented high-resolution crystal structure of LAR D1D2/liprin-alpha3 SAM complex, revealing two conserved key sites involved in binding of LAR-RPTPs with liprin-alpha. Authors next performed cellular experiments to show that liprin-alpha facilitates LAR clustering by inducing oligomerization of liprin-alpha. Authors then found that LAR D1 mediates its homophilic interaction, which is critical for liprin-alpha-mediated LAR clustering and regulation of phosphatase activity. Authors lastly showed that LAR binding to liprin-alpha and liprin-beta is mutually exclusive using various methodologies.

Given the emerging importance of LAR-RPTPs in mediating various aspects of synapse formation and development, I think that this manuscript investigates timely and important topics. In particular, this manuscript (although performed in heterologous cells) addresses unknown roles of LAR-liprin-alpha interaction, which should be further evaluated in followup studies. Overall, data quality is very high and paper was well written. I support the publication of this manuscript in Nature Communications, should the following points are completely and sufficiently addressed:

Major Comments:

1. Quantification of clustering by liprin variants (WT and WQ) and/or LAR mutants (WT, FE, RFN, D1D2, etc) could strengthen the manuscript (Figs.3, 4 and 6).

Following the reviewer's suggestion, we have quantified the LAR clustering data by calculating the percentage of the clustering area per cell. The quantification data are fully consistent with the observations in figures showing the LAR clustering (Figs. 3&4 and Supplementary Figs. 6, 8&10) and have been incorporated into Fig. 3h and Supplementary Figs. 6f, 8g, and 10b in the revised manuscript. In addition, the FA localization of liprin- β 1 shown in Fig. 6f-h were quantified and the quantification data has been incorporated into Supplementary Fig. 12b.

2. Data in panel c of Fig.5 are qualitatively appropriate, but not suitable for quantification as presented

in panel d in the same figure. Did the authors perform this experiment using ECL or I125-based autoradiography? ECL-based immunoblot data are not appropriate for making clear conclusions, given marginal difference in some of group comparisons.

The experiment was performed by using ECL-based immunoblot. Since the LAR clustering located at the ventral-membrane (Supplementary Fig. 7), cell detaching from plates likely disrupts the cluster of LAR and thereby release the clustering-mediated inhibition of the phosphatase activity. Consistent with this notion, the pY signal faded out quickly with time despite the presence of phosphatase inhibitors (Figure I). Although we have minimized the processing time (less than 15 mins), the small amount of unclustered or released LAR might still be able to affect the pY level and flatten the pY signal differences between WT and mutants. Thus, we agree with the reviewer that the immunoblot data is not enough to draw a clear conclusion.

Figure I. Phosphotyrosine (pY) signal detection by using immunoblot. Cells transfected with LAR or its C1548S mutant were lysed and treated with different phosphatase inhibitor cocktails in different concentrations. Immunoblot was performed immediately or 3 hours after cell lysis.

To strengthen our hypothesis that the cluster formation of LAR interferes with its phosphatase activity, we planned to perform an *in vitro* phosphatase assay using the D1-mediated LAR_D1D2 oligomer. Therefore, we designed a double-cysteine mutation in LAR_D1D2 by replacing both L1539 and T1593 to cysteine. As these two residues are close to each other in the D1/D1 interface (Fig. 4b and Supplementary Fig. 9a), the double-cysteine mutation may specifically promote the weak self-association of LAR_D1D2 by forming intermolecular disulfide bridges in solution. Indeed, the double-cysteine mutant (L1539C/T1593C) of LAR_D1D2 formed reversible oligomers in gel filtration column by removing reducing reagents, whereas neither the L1539C nor T1593C mutant showed an obvious oligomer formation (Fig. 4d,e).

Figure 4d,e. (d) Analytical gel filtration analysis showing the oligomer formation induced by the L1539C/T1593C mutation in LAR_D1D2. (e) Molecular weight measurement of different oligomeric fractions of the L1539C/T1593C mutant by using multiangle static light scattering. The theoretical molecular weight of LAR_D1D2 is 65.9 kDa.

We collected the monomer and dimer fractions of the double-cysteine mutant, respectively and measure their phosphatase activities by using a phosphotyrosylpeptide of insulin receptor as substrate (Ramachandran et al. Biochemistry 1992). Based on our structural finding, one of the two active sites in the D1-mediated dimer is blocked while the other remains accessible for reaction. Consistently, the disulfide-linked LAR_D1D2 dimer had ~40 % phosphatase activity of the monomeric form (Fig. 5c). In contrast, when adding 2 mM DTT in the reaction buffer, the phosphatase activity of the LAR_D1D2 dimer was largely recovered to approximately 80 % compared to the monomer fraction (Fig. 5d), presumably due to the dissociation of the disulfide-linked LAR_D1D2 dimer in the reducing environment. Together with the cellular analysis, our data strongly supported that the phosphatase activity of LAR is regulated by the D1-mediated self-association. We have added this part of analysis into the Result section in the revised manuscript.

Figure 5c,d. Phosphatase activity analysis of LAR_D1D2 and its double-cysteine mutant in a reaction buffer without (c) or with DTT (d). The catalytically inactive C1548S (CS) mutant of LAR_D1D2 was used as the negative control.

3. Altered phosphatase activity of LAR induced by interference of D1/D1 interaction is very intriguing notion. Could authors directly test that liprin-alpha WQ mutant compromises the tyrosine phosphorylation level of a subset of LAR substrates (e.g. beta-catenin)?

We thank the reviewer for the comment. With the phosphatase activity assay shown in Fig. 5c,d, we have tried to measure the effects of the oligomerized liprin- α protein and its WQ mutant on the catalytic activity of LAR. However, we failed to obtain the liprin- α proteins containing both the SAM123 and N-terminal oligomerization regions with a reasonable quality by using bacterial, insect cell, and mammalian expression systems. Then, we tried to monitor the pY level of β -catenin by immunoprecipitating endogenous β -catenin of HEK cells overexpressing various constructs of liprin- α 1 and LAR. As shown in Figure II below, the pY signal could be barely detected from the immunoprecipitated β -catenin, presumably due to the quickly decreased pY signal after cell lysis (Figure I). Since the IP experiment plus the immunoblot assay required a processing time much longer than that of the experiment detecting the overall pY level (Fig. 4e,f), probing the pY signal from the IP experiment with the overexpressed LAR phosphatase was difficult for us. Although lacking the direct observation of the compromised pY level of LAR substrates by the WQ mutant of liprin- α , we have employed the phosphatase assay using the previously identified LAR substrate to strengthen the notion that the phosphatase activity of LAR is regulated by the clustering (Fig. 5c,d in the revised manuscript), which are promoted by liprin- α but not its WQ mutant (Fig. 3c,d).

Figure II. Detection of pY level of endogenous β -catenin. Various constructs of LAR were either expressed alone or co-expressed with liprin- α or its WQ mutant in HEK 293T cells. The corresponding position of β -catenin was indicated by a red arrow. The nonspecific protein band appeared in immunoprecipitation was indicated by an asterisk.

4. What is the physiological significance of LAR binding to liprin-beta? To my knowledge, role of liprin-beta has not been seriously investigated, even in heterologous cells. Does liprin-beta similarly induce LAR clustering?

The description of the relationships between LAR, liprin- α , and liprin- β in the original manuscript may be not clear enough. Although liprin- β shares the similar domain organization with liprin- α , LAR has not been found to interact with liprin- β in our and other people's works (Supplementary Fig. 5b, Serra-Pages et al. JBC 1998, and Astigarraga et al. J. Neurosci. 2010). This point has been emphasized in the

revised manuscript to avoid misunderstanding. Rather than promoting LAR clustering, liprin- β may disrupt the liprin- α -promoted clustering of LAR, as the interactions between liprin- α and liprin- β and between liprin- α and LAR are mutually exclusive (Fig. 6b-d). Additionally, as shown in Fig. 6f, liprin- β localized at the peripheral regions of the focal adhesion (FA), implying that liprin- β may be involved in the dynamic regulation of the FA via its interactions with liprin- α and other FA regulators (Sun et al. NCB 2016 and Bouchet et al. eLife 2016).

Supplementary Figure 5b. Analytical gel filtration analysis showing no detectable binding of LAR to liprin- β .

Minor Comments:

1. Typographical errors should be corrected: e.g. line 32, “involved” and line 267 “do not overlapped” are grammarly wrong.

We have carefully checked the whole text and corrected grammar errors in the revised manuscript.

Reviewer #2 (Remarks to the Author):

NCOMMS-19-28108, Xie et al.

Title: Structural Basis of Liprin-alpha-promoted LAR-RPTP Clustering for Modulation of Phosphatase Activity

The Authors report the complex structure of Liprin-alpha3 and the synaptic adhesion protein leukocyte common antigen-related receptor protein tyrosine phosphatase (LAR-RPTP) D1-D2 intracellular domains, and further more suggest a dimeric interaction between the D1 catalytic domains and suggest a model for clustering of the receptors.

The work is of high impact towards understanding the function of RPTPs as presynaptic organizers and role of Liprins in this, and provide lot of interesting new data. Data presented is sound, and text is well written. The paper is suitable for publications in Nat. Communications with the clarifications provided as requested below:

1. Biochemical characterization – SAM 1 and SAM23 alone are shown not to show detectable binding – ITC data on this is lacking? Can you show that, or other direct binding measurement data, as the SEC data is not very quantitative.

Following the reviewer's suggestion, we have performed the ITC-based measurement by titrating SAM1 or SAM23 to LAR_D1D2. Fully consistent with our SEC results, no binding was detected in these measurements (Figure III). Data have been incorporated into Fig. 1f and Supplementary Fig. S3.

Figure III. ITC analysis showing no detectable binding between either SAM1 or SAM23 and LAR_D1D2.

Most importantly the SAM12 construct should be shown also, if this is the deduced functional fragment. What is the affinity of that for the D1D2 construct? I think this would be essential to have here.

We thank the reviewer for the suggestion. Our previous and current structural analysis demonstrated that the three SAM domains together form a structural supramodule (Wei et al. Mol Cell 2011). In particular, SAM2 and SAM3 domains tightly pack with each other and the SAM2/3 interaction is further stabilized by a connecting helix (Figure. IV-a) (Wei et al. Mol Cell 2011). Consistent with the extensive SAM2/3 interdomain interactions, we were not able to obtain the SAM2 or SAM123 fragment with a reasonable quality, suggesting that the folding of SAM2 largely relies on the SAM2/3 interaction. Instead, we used 293T cells to overexpress the truncated liprin- α 1 without the SAM3 domain (Δ SAM3). By using co-IP assay, we found liprin- α 1_ Δ SAM3 lost the binding ability to LAR in co-IP experiment (Figure IV-b). Since the SAM3 domain does not directly participate in the liprin- α /LAR interaction (Fig. 2a), this result supported the importance of the SAM2/3 interdomain interaction in the formation of the structural and functional SAM123 supramodule.

Figure IV. The SAM2/3 interdomain interaction is required for the binding of liprin- α to LAR. (a) Structural analysis showing the tight SAM2/3 interaction in liprin- α 3_SAM123. (b) Co-immunoprecipitation assay showing that liprin- α 1 without the SAM3 domain (Δ SAM3) lost its binding to LAR. “*” indicates cleaved LAR fragment containing a short segment of the extracellular region, the transmembrane helix and the cytoplasmic D1D2 domains (Streuli et al. EMOB J. 1992 and Serra-Pages et al. JBC. 1994).

2. Structure of the complex – the figures should be improved – green and red are not a good combination for colour-blind. I would change the colour-code.

Following the reviewer’s comment, we have changed the representative color of liprin- α to cyan in all related figures.

Fig. 2 stereo figures (c and d) are too small to be seen in stereo (this seems to be overall the problem please enlarge so that they can be viewed (centers of figures should be ca. 6 cm away from each other), also Fig 2e is very small, and need to be larger to provide information.

We thank the reviewer for pointing out the quality issues of the figures. As suggested, we have enlarged Fig. 2e as well as the stereo-views in Fig. 2 and other figures.

It is mentioned that in the asymmetric unit there were four copies of LAR_D1D2 and three copies of SAM123 were “clearly traced”– can the authors explain this in more detail what is meant by this and explain the packing – This is shown in Suppl. Fig 4 but it is not explained that apparently one SAM123 construct is not visible in the maps. Any reasons for this? Also see below.

We agree with the reviewer that our description here was not clearly enough. In the revised manuscript, we have added model-building details and listed the disordered regions of LAR_D1D2 and liprin- α_3 _SAM123 in Supplementary Fig. 4b to clarify this issue. The reviewer also raised an important point that one liprin- α_3 _SAM123 chain is mostly unmodeled due to lacking of clear density. Crystal packing analysis showed that placing a LAR_D1D2-bound SAM123 molecule here, near a crystallographic 2-fold axis, would generate severe clashes between this artificially placed SAM123 molecule and its 2-fold symmetry-related molecule (Supplementary Fig. 4c), thereby preventing the SAM123 molecule from packing stably at this position. We have added the above analysis into the revised manuscript.

Supplementary Figure 4. Structural analysis of the liprin- α_3 _SAM123/LAR_D1D2 complex in crystal. (a) Structural comparison of the LAR_D1D2 structure in the apo form and the α_3 _SAM123 bound form. Four complex structures in one asymmetric unit are overlapped. (b) Composite-omit maps of the four α_3 _SAM123 and LAR_D1D2 complex structures in one asymmetric unit. The 2Fo-Fc densities are contoured in 1.0 σ with the structures superimposed. Corresponding unmodeled regions were listed to the above complex structures. Notably, one α_3 _SAM123 chain (chain G) is mostly unmodeled. (c) Crystal packing analysis. The chain G in crystal was

replaced by a $\alpha 3_SAM123$ molecule (blue) to form a complex with LAR_D1D2 (chain C). The new $\alpha 3_SAM123$ molecule clashes into its symmetric molecule (green). The crystallographic two-fold axis was indicated by an ellipse.

Regarding the binding interfaces – only selected residues are shown. Please report the various interface areas and provide a list of all the contacts e.g. in the supplement.

(for both D1-D1 and SAM123 and Liprin interfaces).

We have reported the D1/D1 and SAM123/D2 interface areas of $\sim 1000 \text{ \AA}^2$ and $\sim 1100 \text{ \AA}^2$, respectively in the Results section and added the full list of the intermolecular contacts in Supplementary Fig. 5a and 9a.

Figure 4. a,b the colours are too pale to properly see, please make the images clearer. Same applies for figure 5, and the supplementary figures with the same grey-pale pink colouring.

We have darkened the ribbon colors in Fig. 4, Fig. 5, and Supplementary Figs. 9&11 in the revised manuscript.

Also please make the electron density figures in Suppl. Figure 4 larger for better visibility, So the trace can be actually seen. The size of the supplementary figures are not particularly restricted? It appears in the figure only SAM1 is shown not SAM123 with e-density although this is mentioned in the legend can you provide please the coordinated and maps for viewing of this? Based on the PDB validation report quite many residues are missing or the fit to density is not very good for the SAM-construct for last two chains. Size of the crystallized and used constructs should be mentioned (number of residues, MW for both proteins and missing residues / regions noted).

Following the reviewer's suggestion, we have enlarged the density figures in Supplementary Fig. 4. The coordinate and map file in pdb and mtz format have been uploaded with the revised manuscript. In addition, we have summarized the protein constructs used in this study for the crystallographic, biochemical, and cellular characterizations (Supplementary Table 1).

Minor points: helices and strands are simply referred to as alpha and beta without explanation e.g. line 13, for clarity please add "sheet" and "helix", and throughout the text. (Again please provide a list of the most important interactions in the text and whole list for each interface in supplement, thank you).

As suggested by the reviewer, we have added "sheet" and "helix" to describe the secondary structural elements throughout the manuscript. We have also presented the important interface residues in Fig. 2c,d, described their interactions in the Results section ("Overall structure of the liprin- $\alpha 3_SAM123/LAR_D1D2$ complex"), and added the full list of the intermolecular contacts in Supplementary Fig. 5a and 9a.

Line 142 "hydrogen bondings" please check for language (hydrogen bonds). Line 144-146 CASK and

Liprin beta interaction are already discussed here but presented in fig. 6. Could you refer to the figure here to make it clearer?

We have corrected the typo and referred the liprin- α /CASK and liprin- α /liprin- β interactions to a new Supplementary Fig. 5c, which is more relevant with the related text and Fig. 2e.

Supplementary Figure 5c. Structural comparison of liprin- α _SAM123 structures in complex with three binding partners respectively.

Stability of D1 and D1-D2 is mentioned, have you tested the stability of these constructs if that really is the cause for the difference in affinity? (e.g. by thermofluor assay), could this be done? Alternatively it maybe the four-fold difference measured is not significant in reality? Is this seen by other methods than ITC? Can you comment?

We have performed the suggested ThermoFluor assay. Consistent with our speculation, the LAR_D1D2 protein showed a higher thermal stability than the D2 domain only (Supplementary Fig. 5d). The previous structural analysis also implicated that the extensive intramolecular D1/D2 interaction limits the flexibility of the D2 domain (Nam et al. Cell 1999). In support of our finding, a recent study showed that the deletion of the D1 domain resulted in of the impaired liprin- α /PTP σ interaction in cells (Bomkamp et al. Front. Synaptic Neurosci. 2019).

Supplementary Figure 5d. Thermal stability analysis of LAR_D1 and LAR_D1D2 by using ThermoFluor assay.

Line 162 and onwards the mutation code is little bit hard to follow so it would be easier if the numbers could be referred to somehow.

The corresponding residue numbers of the mutation sites have been mentioned in the legend of each figure that contains the mutation code. Also, all of the mutations and their codes used in this study have been summarized in Supplementary Table 1.

3. Cluster formation

Mutations nicely show that the D1-D1 interaction is required. It is not very clearly presented though as it is in two different places, and first its said soluble D1D2 failed to generate puncta? But how about in solution? Do you see large oligomers forming?

(This is discussed in discussion perhaps state here if you don't see those and expand the discussion?)

The purified LAR_D1D2 protein was a monomer in gel filtration analysis (Fig. 1b), indicating that the self-association of LAR_D1D2 is extremely weak in solution. It can be explained at least in part by the structural observation that the hydrophilic D1/D1 interface are largely solvent-exposed (Fig. 4b). This point has been added in the Results section ("The clustering of LAR is mediated via the D1/D1 interaction") and also emphasized in the Discussion section.

Overall if possible make clearer the case that membrane attachment is needed, if that appears to be the case, could be in discussed more in discussion.

We thank the reviewer for the suggestion. Our observation showed that deletion of both the extracellular region and the transmembrane helix resulted in the disruption of the liprin- α -promoted clustering of LAR (Fig. 3g). However, we do not have more data to clarify whether the membrane attachment is required for the cluster formation of LAR. Instead, the previous studies indicated that the extracellular region of LAR is required for the cluster formation (Coles et al. Science 2011 and Um et al. Nat Commun 2014), suggesting that our observation is due in part to the deletion of the extracellular region. Considering that this is not the main focus of the paper, we decided not to discuss more about this point.

Also how does the oligomerization propagate beyond the asymmetric unit? Please mention this in the text somewhere.

The D1/D1 interfaces were not found beyond the asymmetric unit. We have added this point in the Results section ("The clustering of LAR is mediated via the D1/D1 interaction").

Line 203 "side-by-side packing fashion" correct the language please.

We have removed "fashion" from the phrase.

Phosphotyrosine assay: is the real ligand for phosphorylation known? Any possibility to follow that?

To our best knowledge, the identify substrates for LAR include insulin receptor (Ramachandran et al. Biochemistry 1992, Kulas et al. JBC 1995), β -catenin (Kypta et al JCB 1996), and EphA2 (Lee et al. Mol Cell Biol. 2013). In our response to the Reviewer 1, we have performed a phosphatase activity assay using an phosphotyrosylpeptide from insulin receptor as substrate to measure the enzymatic activity of LAR_D1D2. Consistent with the partially blocked substrate-binding pocket by the D1/D1 interaction, the crosslinked LAR_D1D2 dimer (Fig. 4d) by introducing a disulfide bond in the D1/D1 interface had significantly reduced phosphatase activity (Fig. 5c).

Line 246 “enzyme-dead”  catalytically inactive.

We have changed the words following the reviewer’s suggestion.

Discussion: see for typos, L308 “ bindings”

We have corrected the typos in the revised manuscript.

Please elaborate on the role of ventral plasma membrane localization here (explain why it is important observation, this is only hinted at, please spell it out clearly).

We thank the reviewer for the suggestion. We have expanded this part of discussion in the revised manuscript.

Please end with a conclusive statement about the current study, not a speculative one which is beyond the current data.

As suggested, we have added a paragraph to conclude our current study at the end of the Discussion part.

REVIEWERS' COMMENTS:

Reviewer #1 (Remarks to the Author):

Revised manuscript was significantly improved by additional data, which could satisfactorily address my previous comments. I recommend the publication of the revised manuscript in Nature Communications, and congratulate authors on this wonderful study.

Reviewer #2 (Remarks to the Author):

I find the corrections well sufficient and support the publication of the revised article in Nature Communications, and thank the authors for thorough revision, just one further note:

Suppl. Fig 4. in a) it should say "superimposed" not "overlapped" - In the same Suppl. fig 4 please add a note about the conclusion on the resulting steric clash e.g. "therefore the fourth SAM123 molecule does not fit in the asymmetric unit in fixed orientation"

- I find this bit intriguing why the molecule does not fit in any particular orientation here.

Apparently there must be some flexibility inherent to the molecule (is there?) or there is something about the crystal packing that is still not fully resolved here. I would have tried to analyse this in closer detail to clarify what is going on - you have the molecules packed around a two-fold but can't fit them - is it the case that only one molecule instead of two can fit here at a time? or just that they can fit in several orientations? (the crystal packing is somewhat complicated due to the large asymmetric unit) Or something else? Could this still be discussed a bit further shortly?

Point-by-point Response

Our responses are shown in blue.

Reviewer #1 (Remarks to the Author):

Revised manuscript was significantly improved by additional data, which could satisfactorily address my previous comments. I recommend the publication of the revised manuscript in Nature Communications, and congratulate authors on this wonderful study.

We thank the two reviewers for their previous constructive comments and suggestions, which helps us to improve our manuscript.

Reviewer #2 (Remarks to the Author):

I find the corrections well sufficient and support the publication of the revised article in Nature Communications, and thank the authors for thorough revision, just one further note:

Suppl. Fig 4. in a) it should say "superimposed" not "overlapped" - In the same Suppl. fig 4 please add a note about the conclusion on the resulting steric clash e.g. "therefore the fourth SAM123 molecule does not fit in the asymmetric unit in fixed orientation"

As suggested by the reviewer, we have revised the figure legend of Supplementary Fig. 4.

- I find this bit intriguing why the molecule does not fit in any particular orientation here. Apparently there must be some flexibility inherent to the molecule (is there?) or there is something about the crystal packing that is still not fully resolved here. I would have tried to analyse this in closer detail to clarify what is going on - you have the molecules packed around a two-fold but can't fit them - is it the case that only one molecule instead of two can fit here at a time? or just that they can fit in several orientations? (the crystal packing is somewhat complicated due to the large asymmetric unit) Or something else? Could this still be discussed a bit further shortly?

We agree with the reviewer that several possibilities might result in such a crystal packing. As the reviewer pointed out, one possibility is that only one SAM123 molecule instead of two can stay near the 2-fold axis. If that was the case, the space group should be *P2* or even *P1*. We have tried to lower the symmetry and reprocess the data. But, we did not observe a better density at the same position by using reprocessed data. Another possibility is that the SAM123 binding to LAR_D1D2 might not be very stable, thereby the SAM123 molecule at this special position partially falling off from the complex. As shown by the B-factor analysis (Fig. A-a), the overall B-factors of SAM123 molecules are higher than those of LAR_D1D2 molecules, supporting the relatively dynamic interaction

between SAM123 and D1D2. In line with the reviewer's speculation, we think that the loosely bound SAM123 at the special position may adopt different orientations in crystal, averaging out the density here. Interestingly, the B-factor analysis also suggests that the D1/D1 crystal packing between LAR_D1D2 molecules is much more stable (Fig. A-b). As this part of analysis is not related to the main theme of the manuscript, we hope that the reviewer would understand our decision not to discuss this issue further.

Figure A. Cartoon and B-putty drawings of the SAM123/D1D2 complex (a) and the crystal packings of D1D2 molecules in one ASU (b).